# The Same–Up–Down Staging System for Recurrent Early Glottic Cancer

**DOI:** 10.3390/cancers15030598

**Published:** 2023-01-18

**Authors:** Giuseppe Licci, Luca Giovanni Locatello, Giandomenico Maggiore, Flavia Cozzolino, Saverio Caini, Oreste Gallo

**Affiliations:** 1Department of Otorhinolaryngology, Careggi University Hospital, Largo Brambilla 3, 50134 Florence, Italy; 2Department of Otorhinolaryngology, Academic Hospital “Santa Maria della Misericordia”, Azienda Sanitaria Universitaria Friuli Centrale, Piazzale Santa Maria della Misericordia 15, 33100 Udine, Italy; 3Cancer Risk Factors and Lifestyle Epidemiology Unit, Institute for Cancer Research, Prevention and Clinical Network (ISPRO), Via Cosimo il Vecchio 2, 50139 Florence, Italy; 4Department of Experimental and Clinical Medicine, University of Florence, Viale Morgagni 48, 50134 Florence, Italy

**Keywords:** laryngeal cancer, glottic cancer, recurrence, staging, squamous-cell carcinoma

## Abstract

**Simple Summary:**

Compared with other head and neck cancer types, the prognosis of recurrent early glottic cancer (rEGC) may be less dependent on the presence of regional (N) and/or distant metastases (M). The latter two are clinically infrequent due to rEGC’s peculiar biology, but the currently available staging systems still rely upon these parameters. Thus, we developed a new staging system (SUD) centered on the comparison between the T stages of the recurrence and the primary tumor. Then, in our cohort of 258 patients with rEGC treated at our Institution in Florence, Italy, we verified how the SUD system performs in the prediction of the overall and disease-specific survival, compared to the other classifications already in use.

**Abstract:**

(1) Background: The treatment of recurrent early glottic cancer (rEGC) remains challenging. We wanted to investigate how the oncological outcomes are affected by the initial and recurrent stages, in order to propose our newly developed Same–Up–Down (SUD) staging system. (2) Methods: In our cohort of 258 rEGC patients, we retrospectively assessed the prognostic performances of the rTNM (the TNM staging system for recurrence), CLRSS, CLRSS-2, and SUD staging systems by univariate and multivariate Cox analysis, comparing their predictive capability using Harrell’s C-index. (3) Results: The SUD classification satisfactorily predicted both overall survival (*p* = 0.022) and second-recurrence-free survival (*p* = 0.024, as same + down vs. upstage) in our cohort. It also outperformed the other three systems in terms of prediction of survival, with an improvement of 1.52%, 1.18%, and 3.96% in the predictive capacity of overall survival, disease-specific survival, and second-recurrence-free survival, respectively. (4) Conclusions: The SUD staging system can efficiently predict survival in rEGC patients, whose prognosis heavily depends on both the initial and recurrent locoregional extension.

## 1. Introduction

Early glottic cancer (EGC) is commonly defined as a squamous-cell carcinoma that does not extend beyond the true vocal cords or impair their motion, and without any clinical or radiological evidence of cervical node involvement [1]. According to the eighth edition of the American Joint Committee on Cancer and Union for International Cancer Control (AJCC/UICC)’s tumor–node–metastasis (TNM) system, EGC corresponds to the T1aN0M0, T1bN0M0 (stage I), and T2N0M0 (stage II) lesions [2]. Oncological outcomes for EGC are known to be excellent, with reported 5-year disease-specific survival (DSS) of over 80–90% [1,3,4]. Exclusive radiotherapy (RT) or surgery, mostly in the form of transoral laser microsurgery (TLM), are the two main primary treatment options, with comparable results in terms of survival even in the most recent series [1,5]. With the ultimate aim of optimizing the functional outcomes, TLM is performed in the most conservative way, because tumor-free margins of one millimeter are usually sufficient and lead to the voice recovering faster [6,7]. On the other hand, new RT techniques are being investigated, such as single vocal cord irradiation (with apparently good results) [8] or the use of stereotactic RT, which is still in its preliminary phase [9,10].

Unfortunately, around 10–20% of EGC cases still relapse and need salvage treatment [1,11]. The management of recurrent EGC (rEGC) depends on many factors, including the initial treatment, the patient’s general health status, the surgical expertise available, and the clinical/pathological stage [1]. Some years ago, in addition to the classical rTNM classification, some authors proposed a specific staging system for laryngeal carcinoma—the composite laryngeal recurrence staging system (CLRSS), and its updated version CLRSS-2 [12,13]. These systems, however, were built upon two cohorts of all-stage recurrent laryngeal cancer, while for rEGC—whose risk of regional and distant metastases is notably low [1,14]—no specific tool exists to the best of our knowledge. In the present paper, we aim to assess the predictive power of the currently available staging systems and explore a novel classification that can stratify the oncological outcomes for patients who have already had a relapse of the disease.

## 2. Materials and Methods

### 2.1. Study Population

From our institutional database of patients diagnosed with laryngeal cancer between January 1980 and December 2021, we retrospectively extracted patients who experienced a locoregional recurrence after initial treatment for early glottic cancer, collecting their clinical and pathological information, such as age at first diagnosis, sex, smoking and alcohol abuse status, initial TNM stage at diagnosis, type of initial treatment (e.g., radiotherapy, transoral laser surgery, partial or total laryngectomy), disease-free interval, follow-up period, and cause of death. For each recurrence, we retrieved the rTNM, CLRSS, and CLRSS-2 stages, the chosen salvage treatment, distant metastases, and disease-free interval.

We included patients with an initial early glottic cancer (T1-T2 patients according to the 8th edition of the American Joint Committee on Cancer (AJCC), without lymph node involvement) [15] who had experienced recurrent disease after a disease-free interval of at least 3 months, and with a histological diagnosis of squamous-cell carcinoma. We excluded non-squamous-cell histology, patients who underwent a palliative treatment without curative intent, and patients with incomplete data.

The follow-up protocol, both for primary tumors and recurrences, was conducted by the medical staff (seniors and residents in otolaryngology) of our department, and it usually consisted of direct/indirect laryngoscopy every 4–8 weeks for the first 2 years, every 3 months for the 3rd year, every 6 months during the 4th and 5th years, and then once a year.

### 2.2. Restaging Systems Used for the Classification of Tumor Recurrence

The recurrences were initially staged using three different staging systems: rTNM [15], CLRSS [12], and CLRSS-2 [13]. Then, we retrospectively applied a newly developed three-tiered classification called the Same–Up–Down (SUD) staging system to our cohort. In brief, the TNM stages of recurrences were compared with those of primary tumors. If the recurrence had a higher TNM stage than the primary tumor, it was classified as “up-stage”, if it had a lower TNM stage it was classified as “down-stage”, and if the two stages were the same the recurrence was classified as “same-stage”.

### 2.3. Statistical Analyses

We used the Kaplan–Meier method to describe the overall, disease-specific, and second-recurrence-free survival with complete follow-up, or with follow-up censored at 5 years and 10 years since primary recurrence. We applied the log-rank test to compare the survival function across categories of several patients’ demographic and tumor characteristics, including age at first recurrence, sex, cigarette smoking and alcohol intake habits, initial treatment, tumor stage at diagnosis, salvage at first recurrence (radiotherapy, transoral laser microsurgery, partial laryngectomy, and total laryngectomy), selective neck dissection, presence of distant metastases, and each of the four staging systems. We fitted univariate and multivariate Cox regression models to quantify the association of the aforementioned patients’ demographic and tumor characteristics with the hazard of death, disease-specific death, and second tumor recurrence. We then used Harrell’s C-index [16] to establish whether the SUD staging system had a better predictive capability than the systems currently in use (i.e., r-TNM, CLRSS, and CLRSS-2).

Given the small number of down-stage patients (*n* = 14), the definitive analyses for the SUD staging system were carried out by comparing up-stage vs. down/same-stage patients (merged into a single group and taken as a reference). For the same reason, we merged stages 0 and I and stages III and IV of the CLRSS and CLRSS-2 classifications into single categories, as well as patients coded as 0 and I according to the rTNM classification.

In order to validate our results, we randomly split the study sample into a training set and a validation set, each accounting for 50% of the original study sample. We then fitted the model in the training set, and the model thus obtained was then used in the validation test to assess whether adding the SUD variable to the model would bring any improvement in its predictive ability (as quantified by means of Harrell’s C-index statistics). This was repeated for the three endpoints under study, i.e., OS, DSS, and SRFS.

The statistical analyses were performed with Stata software (StataCorp, 2015, Stata Statistical Software Version 14). All statistical tests were two-sided, and a *p*-value of less than 0.05 was considered statistically significant.

## 3. Results

A cohort of 258 patients was ultimately analyzed after applying the selection criteria, and its general description is displayed in Table 1. Regarding primary tumors, 182 (70.5%) patients were classified as TNM stage I; in particular, 69 (26.7%) patients were classified as T1a, 113 (43.8%) as T1b, and 76 patients (29.9%) as stage II. A total of 70 (27.1%) patients had an initial surgical treatment, whereas 188 (72.9%) underwent primary radiotherapy. A total of 101 (39.1%) patients developed their first recurrence within 12 months from the completion of the primary treatment, and 237 (91.9%) within 5 years, while the other 21 patients developed a recurrence/second primary laryngeal tumor after 5 years. By using our system, we identified 99 (38.4%) same-stage, 145 (56.2%) up-stage, and 14 (5.4%) down-stage cases. As salvage therapy, 49 (19.0%) patients received transoral laser microsurgery, 31 (12.0%) had open partial laryngectomy, 171 (66.3%) underwent total laryngectomy, and 7 (2.7%) received radiotherapy. Out of the total of 258 rEGC patients in this study, 67 (26.0%) had a second recurrence. Among patients with primary stage I, those with an up-stage recurrence had OS and DFS rates of 65.1% and 66.1%, respectively, over the observation period, while the same group with same- or down-stage recurrence had better survival rates (83.6% and 83.6% for OS and DFS, respectively).

In contrast, among patients with initial stage II, those with down-stage recurrence had an OS and DFS of 83.3% and 100%, respectively, while the same group with same- or up-stage relapse had worse OS and DFS: 62.5% and 71.9%, respectively.

Within the class of up-stage relapses, patients with a “more-than-one-step up-stage” (for example, from stage I to III) presented a lower mean SRFS (47 months, 67 patients) compared to one-step (e.g., from stage I to II) up-stage patients (78 patients with a mean SRFS of 55 months).

According to the survival regression models, age at first recurrence was predictably associated with worse overall survival (OS) on multivariate analysis (HR = 2.33, *p* = 0.006), but also with worse DSS (HR= 2.55, *p* = 0.032). Other independent predictors of OS included stage at diagnosis (HR = 1.71, *p* = 0.047), SUD (HR = 1.86, *p* = 0.022), second locoregional recurrence (HR = 4.06, *p* ≤ 0.001), and distant metastases (HR = 7.78, *p* < 0.001). For DSS, other significant factors that emerged from our analysis were age > 71 years (HR = 2.55 *p* = 0.032), smoking habit (HR = 3.18, *p* = 0.003), CLRSS stage II (HR = 3.87, *p* = 0.001), dichotomous SUD (HR = 2.96, *p* = 0.012), salvage RT (HR = 18.95, *p* = 0.007), second locoregional recurrence (HR = 33.83, *p* ≤ 0.001), and distant metastasis (HR = 23.99, *p* < 0.001). Finally, the variables independently associated with second-recurrence-free survival were initial treatment with surgery as a protective factor (HR = 0.26, *p* = 0.001), CLRSS stage II (HR = 2.21, *p* = 0.020), and dichotomous SUD (HR = 1.92, *p* = 0.024).

By univariate analysis, we found other significances for the female sex as a risk factor for a worse OS and DSS, primary surgery as a protective factor for OS and second-recurrence-free survival (SRFS), salvage TLM as a protective factor for OS and DSS, and stage at first recurrence predicting OS, SRFS, and DSS. All of the analyses for the significant factors in predicting OS, second-recurrence-free survival, and disease-specific survival are presented in Table 2.

Kaplan–Meier curves of the overall survival, disease-specific survival, and second-recurrence-free survival for the SUD, rTNM stage, CLRSS, and CLRSS-2 classes are displayed in Figure 1, Figure 2 and Figure 3, respectively. Table 3 shows how the SUD system appears to significantly stratify our cohort for both 5- and 10-year OS and second-recurrence-free survival by multiple Cox regression analysis; in particular, this system was able to predict both OS (*p* = 0.022, considering the total follow-up time) and SRFS (*p* = 0.024) when a dichotomous classification (same + down versus upstage) was implemented. Similar results were obtained when considering the 10-year (*p* = 0.016 for OS and *p* = 0.027 for RFS) and 5-year follow-up times (*p* = 0.040 for OS and *p* = 0.027 for RFS). Lastly, when evaluating the different staging systems with Harrell’s C-index analysis, the addition of the SUD variable was correlated with an improvement of 1.52%, 1.18%, and 3.96% in the predictive capacity of overall survival, disease-specific survival, and second-recurrence-free survival, respectively, compared to other staging methods alone. Finally, when adding the SUD variable to the model, Harrell’s C-index in the validation cohort increased by 2.1% for OS, 0.5% for DFS, and 3.8% for SRFS.

## 4. Discussion

Early-stage glottic squamous-cell carcinoma portends a very favorable prognosis compared to other types of head and neck cancer [1]. An aggressive approach to persistent hoarseness/dysphonia has been promoted since the previous century [17], and it has been recently shown how an early referral to an otolaryngologist, as measured by the dysphonia-to-diagnosis interval, is even associated with improved survival [18]. A single-stage treatment based on surgical resection or definitive RT on the larynx is the best therapeutic option in terms of both survival and functional preservation, while no elective treatment of the neck’s lymph nodes is necessary, as recently stressed in an analysis by the US National Cancer Database [19].

Relapse rates are usually low, and they are not significantly different between EGC treated with RT (estimated risk at 3 years of 8.7%, with a 95% CI of 5.6–12.7%) or with TLM (risk of 8.7%, 95% CI of 5.9–12.3%) [4], although it has been shown how the initial RT treatment may be associated with worse outcomes in terms of survival [20].

In our analysis, age and female gender (the latter in univariate analysis only) were correlated with worse OS and DSS. The roles of age and sex are already known factors in the literature on primary early glottic cancer; for instance, Nomura et al. recently showed how advanced age in patients with primary tumors is associated with worse outcomes, while the association with sex is more variable [21]. Evidence is lower for recurrent tumors. Haapaniemi et al. [22] identified only the female sex with locoregional relapse, but not older age. It could be possible that women, in whom glottic carcinoma is much less frequent, may present a more aggressive disease, or a greater likelihood of refusing to undergo treatments that are functionally and aesthetically more disabling [23]. This would also explain why 11 out of the 12 women in our database initially underwent radiation therapy. Unfortunately, our analysis on gender is weak due to the low number of female patients included. Since only univariate analysis was performed, there may be other confounding factors, such as smoking (only one woman was a non-smoker—8.3% compared to 43.5% of men), while the mean age between the two sexes was similar (66 years for women, 67 for men, *p*-value non-significant by *t*-test). Nonetheless, the information on laryngeal cancer in women remains very poor and even outdated [24,25].

Concerning smoking, it was unsurprisingly associated with worse disease-specific survival in our group with recurrent early glottic cancer [26]. It would have been interesting to verify the abstention or continuation of the smoking habit even after the diagnosis in our cohort, but we did not retrieve such data from the medical records. In fact, it has been shown that cancer survivors who continue to smoke are less likely to respond to treatment, show higher toxicity under chemotherapy, and have lower survival rates than patients who quit smoking before or at the time of diagnosis [27].

Unfortunately, a very large proportion of rEGC cases are salvaged by total laryngectomy (74% of cases in a large series from the Memorial Sloan Kettering Cancer Center, where 72% had early-stage disease) [28]. For recurrent cases, even in an irradiated field, organ-preservation strategies such as TLM or open partial laryngeal surgery are often feasible [1,29], and they do not compromise survival in comparison with TL [30,31]. Our findings do not reveal the same for salvage radiotherapy, which appears to worsen disease-free survival. The use of salvage radiotherapy after the failure of a surgical strategy is not a very frequent topic in the literature [1]; however, Akbaba et al. have published their experience with very promising results of salvage RT, discussing how in other cases [32] salvage surgery has led to better overall survival than salvage RT due to many biases, including the initial stage of the tumor [33]. This is in contrast to our study, which considered only tumors that were initially EGC and in which salvage RT had less satisfactory results.

Another result derived from our analysis is that salvage TLM acts as a protecting factor for overall survival in univariate analysis, in comparison to the other open surgeries. This may be explained by the fact that the candidates for this type of surgery usually present with a less advanced stage of relapse. In this regard, Piazza et al. have recently published indications on the use of TLM for post-RT relapses. Specifically, the authors recommend it only for the less advanced stages of the disease (up to rT2), with optimal laryngeal exposure (e.g., an adequate “laryngoscore”). Furthermore, the initial tumor stage must also be considered in the decision-making process, due to the risk that primary T2 tumors show frequent tumoral foci interspersed in the cicatricial tissues [34].

This last indication, as well as the influence of the initial stage of disease on the efficacy of salvage RT discussed above, suggests the fundamental role of the initial stage of disease in a staging system for relapses, as already shown for the CRLSS system in 1998 [12].

Regarding the comparison of the three staging systems used in this paper, using Harrell’s C-index, a 1.52% increase in the predictive capacity of our system for OS should be regarded as relevant if we consider that survival also depends on other factors than SUD, and which this tool does not account for. The 1.18% increase in predictive capacity for DSS is also a remarkable value considering that the starting predictive capacity is already very high (91%). In fact, the currently available restaging systems (i.e., rTNM and the two versions of the CLRSS) appear to be more effectively implemented after the treatment of initially advanced tumors, where the risk of nodal and distant metastasis is notably high. This could be why in our case the CLRSS and CLRSS-2 staging systems showed worse performance in stage II and even more so in stages III and IV (as also seen in Figure 1, Figure 2 and Figure 3), in which the nodal involvement plays a significant role in the determination of survival. In the specific case of early glottic cancer recurrences, we can see how our staging system performs better than the CLRSS systems, and this demonstrates how the prognosis is dependent almost exclusively on the local extension. It must be taken into account that stages III and IV of CLRSS and CLRSS II were merged due to the small sample size. In Table 2, it is possible to note that the hazard ratio for TNM stage II recurrences is greater than that for the up-stage, suggesting that the rTNM stage was better for predicting worse outcomes. However, it must be considered that the SUD system acts on all levels of the TNM and is intended for use as a prognostic tool alongside it—not to replace it. Therefore, even though the already present staging systems can adequately predict OS and DSS, the addition of the SUD system for early glottic cancers may help to choose a different therapeutic alternative. For example, there are scenarios where, in face of the same rTNM, a more extensive resection may be required (e.g., salvage total laryngectomy and not partial laryngectomy) due to increased risk of poor disease control if the recurrence results are up-stage rather than same- or down-stage.

The new SUD staging system also seems to have room for improvement by further stratifying the classes. We noticed that the “more-than-one-step up-staged” patients presented a lower mean SRFS compared to “one-step up-staged” patients. This could further develop the SUD staging system by means of a more complete dataset, perhaps even with a division of down-stage patients—too few in our dataset—into “one-step” or “more-than-one-step” down-stage patients.

Limitations of our study include the long timespan considered, even though stratified analysis for patients treated in the early versus the late period did not reveal any differences. For example, in our patient cohort, we noticed a greater use of radiotherapy in the first 20 years compared to the last 20 years, when TLM became more common as a treatment for primary T1–T2 tumors. Another issue of our analysis is represented by the evaluation of surgical margins, since they could affect disease-free survival. At our department, only since 2005 have we begun to send the excised TLM specimens for intraoperative frozen sections, with subsequent surgical enlargement in case of R1. Before that, positive resection margins on the final histological examination could be corrected only with a second treatment stage. Then, as a verification of the accuracy of the follow-up protocols, we conducted a Student’s *t*-test analysis comparing the disease-free interval between the primary tumor and recurrence in the rT1–rT2 and rT3–rT4 groups. The lack of significance that we found (*p* = 0.735), at minimum, indicates the correct execution of the surveillance protocol, and led us to conclude that the recurrences at an advanced stage were not likely due to missed follow-up visits.

Furthermore, while our system is specifically devised for rEGC, our cohort was mainly composed of patients who were previously irradiated; therefore, a further stratification according to the initial treatment could be introduced.

Another limitation of this study is the presence of a few down-stage patients in our dataset, which prevented the evaluations from being carried out for each of the classes of the SUD system; therefore, we had to merge the same-stage and down-stage patients in a single group for the analyses.

The reliability of our results was reinforced by the observation that adding the SUD variable to the model would also bring an improvement in its predictive ability upon splitting the study sample into a training set and a validation set. However, this should still be considered a preliminary step, and confirmation in an independent cohort of patients treated at another hospital (and possibly differing from our study sample in terms of demographics and clinical characteristics) is required.

## 5. Conclusions

The prognosis of rEGC depends mostly on local control of the disease, making the tumor stage the most important factor to consider in the salvage setting. We have developed an rEGC-specific staging system that outperforms the results of other currently available classifications, and we plan to validate it in future prospective and multicenter studies.

## Figures and Tables

**Figure 1 cancers-15-00598-f001:**
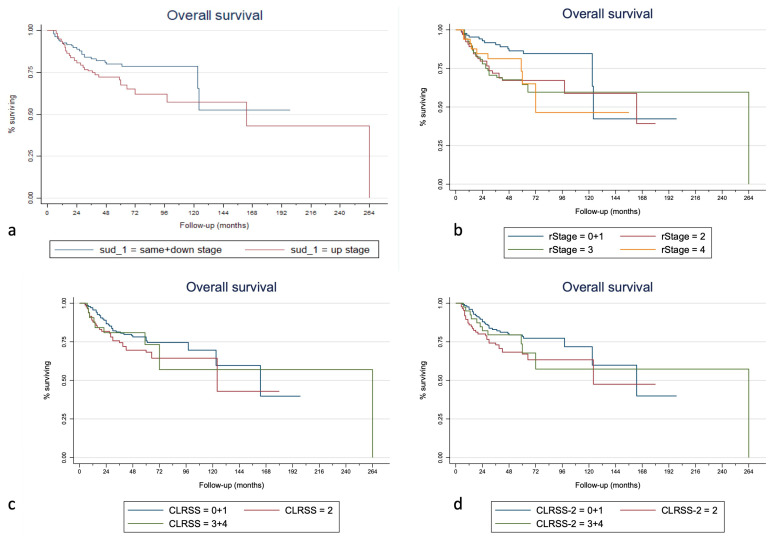
Kaplan–Meier curves for overall survival stratified for the SUD (**a**), rTNM (**b**), CLRSS (**c**), and CLRSS-2 (**d**) staging classes (merged as the analyses were performed).

**Figure 2 cancers-15-00598-f002:**
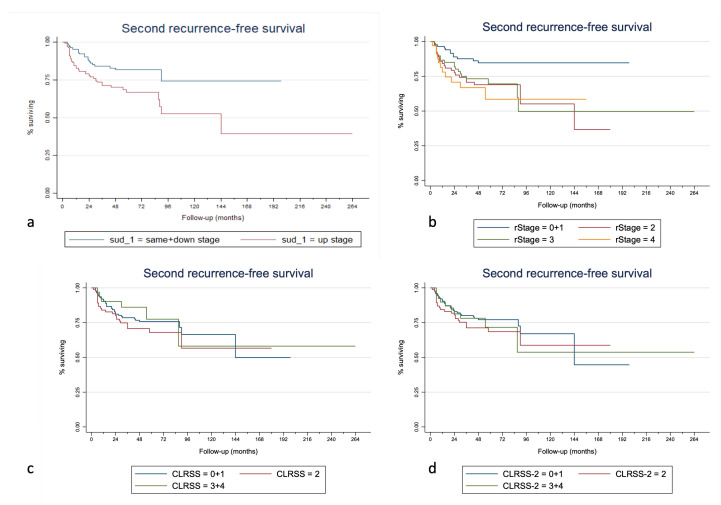
Kaplan–Meier curves for second-recurrence-free survival stratified for the SUD (**a**), rTNM (**b**), CLRSS (**c**), and CLRSS-2 (**d**) staging classes (merged as the analyses were performed).

**Figure 3 cancers-15-00598-f003:**
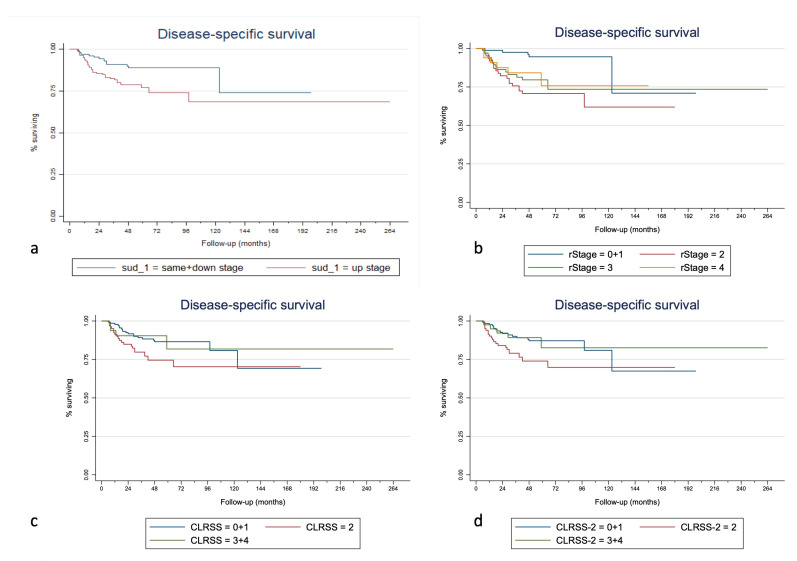
Kaplan–Meier curves for disease-specific survival stratified for the SUD (**a**), rTNM (**b**), CLRSS (**c**), and CLRSS-2 (**d**) staging classes (merged as the analyses were performed).

**Table 1 cancers-15-00598-t001:** A general descriptive analysis of the variables analyzed in our population.

		Frequency	Percentage
Gender	Male	246	95.3
	Female	12	4.7
Smoking status	No	108	41.9
	Yes	150	58.1
Alcohol abuse	No	206	79.8
	Yes	52	20.2
Original treatment	Surgery	70	27.1
	Radiotherapy	188	72.9
Original TNM stage	I	182	70.5
	II	76	29.5
Recurrence TNM stage	I	89	34.5
	II	67	26.0
	III	68	26.4
	IV	34	13.2
CLRSS	I	140	54.3
	II	85	32.9
	III	15	5.8
	IV	18	7.0
CLRSS-2	I	129	50.4
	II	88	34.1
	III	22	8.5
	IV	19	7.4
Same–up–down stage	Same	99	38.4
	Up	145	56.2
	Down	14	5.4
Salvage treatment	TLM	49	19.0
	PL	31	12.0
	TL	171	66.3
	RT	7	2.7
Distant metastases	yes	10	3.9
	no	248	96.1
TOTAL		258	100

Acronyms: TLM = transoral laser microsurgery; PL = partial laryngectomy; TL = total laryngectomy; RT = radiotherapy.

**Table 2 cancers-15-00598-t002:** Log-rank test, along with univariate and bivariate Cox regression analyses, for overall survival, second-recurrence-free survival, and disease-specific survival for each considered factor; * denotes a *p*-value < 0.05. Acronyms: S = same-stage; U = up-stage; D = down-stage; PL = partial laryngectomy; TL = total laryngectomy; TLM = transoral laser microsurgery; RT = radiotherapy.

Variable		Overall Survival					Second Recurrence-Free Survival						Disease-Specific Survival			
		*p*-Value Log-Rank Test	Univariate Cox Regression		Multiple Cox Regression		*p*-Value Log-Rank Test	Univariate Cox Regression		Multiple Cox Regression		*p*-Value Log-Rank Test	Univariate Cox Regression		Multiple Cox Regression	
			HR	*p*-Value	HR	*p*-Value		HR	*p*-Value	HR	*p*-Value		HR	*p*-Value	HR	*p*-Value
Age at first	<60		1.00		1.00			1.00		1.00			1.00		1.00	
recurrence	61–70		1.23	0.519	1.53	0.191		0.69	0.216	0.71	0.251		0.74	0.449	1.80	0.176
	71+	0.368	1.50	0.166	2.33	0.006 *	0.093	0.53	0.035 *	0.65	0.155	0.648	1.03	0.923	2.55	0.032 *
Sex	male		1.00					1.00					1.00			
	female	0.029 *	2.32	0.034 *			0.536	1.37	0.540			0.004 *	3.33	0.006 *		
Smoking	no		1.00		1.00			1.00					1.00		1.00	
	yes	0.304	1.27	0.307	1.51	0.095	0.994	1.00	0.994			0.114	1.64	0.118	3.18	0.003 *
Alcool	no		1.00					1.00		1.00			1.00			
	yes	0.804	1.07	0.804			0.414	1.26	0.417	1.20	0.538	0.830	1.08	0.830		
Initial	rt		1.00		1.00			1.00		1.00			1.00		1.00	
treatment	surgery	0.005 *	0.40	0.007 *	0.62	0.216	0.002 *	0.33	0.003 *	0.26	0.001 *	0.056	0.46	0.063	0.54	0.230
Stage at	1		1.00		1.00			1.00		1.00			1.00			
diagnosis	2	0.393	1.23	0.396	1.71	0.047 *	0.665	0.89	0.667	0.54	0.099	0.788	1.09	0.789		
Stage at first	0 + 1		1.00					1.00					1.00			
recurrence	2		2.30	0.014 *				2.51	0.011 *				5.42	0.001 *		
	3		2.56	0.005 *				2.39	0.016 *				4.15	0.006 *		
	4	0.018 *	2.57	0.017 *			0.013 *	3.27	0.004 *			0.003 *	4.10	0.016 *		
CLRSS	0 + 1		1.00					1.00		1.00			1.00		1.00	
	2		1.44	0.147				1.35	0.247	2.21	0.020 *		2.07	0.021 *	3.87	0.001 *
	3 + 4	0.339	1.23	0.557			0.411	0.88	0.766	1.39	0.481	0.055	1.20	0.719	3.73	0.093
CLRSS-2	0 + 1		1.00					1.00					1.00			
	2		1.66	0.048 *				1.37	0.236				2.18	0.015 *		
	3 + 4	0.116	1.51	0.202			0.486	1.20	0.611			0.035 *	1.16	0.749		
SUD	S + D		1.00		1.00			1.00		1.00			1.00		1.00	
(complete follow-up)	U	0.029 *	1.70	0.032 *	1.86	0.022 *	0.005 *	2.09	0.007 *	1.92	0.024 *	0.012 *	2.26	0.015 *	2.96	0.012 *
Salvage at	PL		1.00					1.00					1.00		1.00	
first	TL		1.30	0.480				0.84	0.604				0.98	0.972	1.10	0.852
recurrence	RT		3.97	0.084				-	-				4.85	0.055	18.95	0.007 *
	TLM	0.003 *	0.29	0.041 *			0.328	0.49	0.133			0.007 *	0.28	0.075	1.23	0.800
Elective	no		1.00		1.00			1.00					1.00		1.00	
neck dissection	yes	0.460	0.80	0.463	0.65	0.219	0.517	1.21	0.520			0.595	0.81	0.597	0.34	0.062
Second	no		1.00		1.00								1.00		1.00	
locoregional recurrence	yes	<0.001 *	3.59	<0.001 *	4.06	<0.001 *						<0.001 *	14.75	<0.001 *	33.83	<0.001 *
Distant	no		1.00		1.00								1.00		1.00	
metastases	yes	<0.001 *	8.39	<0.001 *	7.78	<0.001 *						<0.001 *	14.45	<0.001 *	23.99	<0.001 *

**Table 3 cancers-15-00598-t003:** Log-rank test, along with univariate and bivariate Cox regression analyses, for overall survival, second-recurrence-free survival, and disease-specific survival on SUD; * denotes a *p*-value < 0.05. Acronyms: SUD-c = SUD on complete follow-up; SUD60 = SUD on follow-up of 60 months; SUD120 = SUD on follow up of 120 months; S = same-stage; U = up-stage; D = down-stage.

Variable		Overall Survival					Second-Recurrence-Free Survival				
		*p*-Value Log-Rank Test	Univariate Cox Regression		Multiple Cox Regression		*p*-Value Log-Rank Test	Univariate Cox Regression		Multiple Cox Regression	
			HR	*p*-Value	HR	*p*-Value		HR	*p*-Value	HR	*p*-Value
SUD-c	S		1.00					1.00			
	U		1.60	0.065				1.79	0.031 *		
	D	0.075	0.57	0.447			0.007 *	-	-		
SUD-60	S		1.00					1.00			
	U		1.40	0.218				1.78	0.033 *		
	D	0.159	0.32	0.227			0.008 *	-	-		
SUD-120	S		1.00					1.00			
	U		1.66	0.053				1.78	0.034 *		
	D	0.032 *	0.30	0.237			0.007 *	-	-		
SUD-c	S + D		1.00		1.00			1.00		1.00	
	U	0.029 *	1.70	0.032 *	1.86	0.022 *	0.005 *	2.09	0.007 *	1.92	0.024 *
SUD-60	S + D		1.00		1.00			1.00		1.00	
	U	0.105	1.54	0.108	1.82	0.040 *	0.006 *	2.08	0.007 *	1.91	0.026 *
SUD-120	S + D		1.00		1.00			1.00		1.00	
	U	0.016 *	1.83	0.018 *	1.96	0.016 *	0.005 *	2.09	0.007 *	1.90	0.027 *

## Data Availability

The data presented in this study are available in this article. Further data are available upon reasonable request to the corresponding author.

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
