# Peer review of "The Same–Up–Down Staging System for Recurrent Early Glottic Cancer"

_cancers, 2023, doi:10.3390/cancers15030598_

Round 1

Reviewer 1 Report

The Authors in their contribution describe a novel staging system, the Same-Up-Down (SUD) staging, for recurrent early glottic cancer (rEGC) that is based on comparison between the T stages of the recurrent tumor and the primary tumor. They compare the prognostic potential of SUD with more traditional rTNM, CLRSS and CLRSS-2 staging systems and show improved prediction of patient survival, albeit by a small margin. The article is well written, easy to read and provides clear results with possible translation to clinical application. However, the article has weaknesses that should be addressed before its publication. Major: i) The main weakness is that the presented results are not validated in an independent dataset. Without proper validation the SUD staging cannot be used as a general staging system for different clinical centres. I would suggest the Authors to reuse the data from the references [12] and [13], which should be available, or any other independent dataset. ii) Low number of women included in the study may lead to misinterpretation of the results of univariate analyses. For example, the Authors state that in univariate analysis "sex" is a risk factor for OS and DSS, however, they do not see the same risk in multivariate analysis. It very well may be that 14 women included in the study were much older than men, or were more often smokers and the putative risk factor "sex" may only hide these primary risk factors. From the provided tables, I was not able to resolve this issue. The Authors should discuss this issue, add contingency table for "sex" and other risk factors or restrict the analysis on men only. iii) Similarly, in the SUD staging, there were only 14 cases of patients with "Down" stage making it difficult to interpret its risk factor. Minor: iv) Please clearly define meaning of rTNM abbreviation. v) Figures 1 to 3: It would be interesting to see the performance of other staging systems along with the proposed SUD system, e.g., by replacing the first panel of each Figure with three panels for rTNM, CLRSS, and CLRSS-2 or by providing supplementary figures. —

Reviewer 2 Report

Authors describe a new staging system for glottic cancer and evaluate this system in their cohort. Authors address the strengths and weaknesses of their analysis. 

Major comment

A new classification system is proposed, interpretation of the results and impact of the conclusions would benefit from a rEGC validation cohort. If unavailable (in literature of through collaboration) is the classification a robust when the author cohort is randomly (avoiding the influence of time, gender etc) split in two? )

Minor comments: the manuscript would benefit from proofreading by a native speaker. The tables could use some reformatting (perhaps reduce the row hight? and transpose the table or page? and the KM curves could be presented in a "matrix"like figure?). 

Reviewer 3 Report

1. In Table 1, for each characteristic, the authors repeat the line Total. Since this line is the same everywhere, I suggest removing it so as not to overload the table.

2. The need to introduce an additional SUD criterion is not entirely clear. Table 2 shows survival by stage at the first recurrence. It can be seen that for stages other than 0+1, survival rates decrease, with a higher hazard ratio than for SUD. In this regard, stage 2 and above at the first relapse is an unfavorable factor. Another thing is interesting: in what proportion of patients with initially stage 0 + 1 did the stage increase at the first relapse and this led to an unfavorable outcome? And vice versa, initially the second stage at the first recurrence was 0 + 1 and the outcome is favorable? And if you add to your SUD criterion an additional division with U= up-stage into those whose stage has grown by 1 (for example, 0 + 1 to 2) and those who have grown more (2 to 4 for example)? then in the second case the forecast will be worse? Thus, it seems to me that a more detailed analysis of the possibilities of the SUD criterion introduced by the authors is needed.

Round 2

Reviewer 1 Report

Dear Authors, You have addressed all my comments in satisfactory way. Thank you.

Author Response

revised

Reviewer 3 Report

I have no more remarks/comments to the authors of the article. I believe that in its present form the manuscript can be recommended for publication. There are some minor comments to table 2 and figures, but they will be corrected by the authors when editing the manuscript.

Author Response

revised